# Enhancement of Neuroblastoma NK-Cell-Mediated Lysis through NF-kB p65 Subunit-Induced Expression of FAS and PVR, the Loss of Which Is Associated with Poor Patient Outcome

**DOI:** 10.3390/cancers13174368

**Published:** 2021-08-29

**Authors:** Elisa Brandetti, Chiara Focaccetti, Annalisa Pezzolo, Marzia Ognibene, Valentina Folgiero, Nicola Cotugno, Monica Benvenuto, Paolo Palma, Vittorio Manzari, Paolo Rossi, Doriana Fruci, Roberto Bei, Loredana Cifaldi

**Affiliations:** 1Academic Department of Pediatrics (DPUO), Ospedale Pediatrico Bambino Gesù, IRCCS, 00165 Rome, Italy; e.brandetti@hotmail.it (E.B.); paolo.rossi@opbg.net (P.R.); 2Department of Human Science and Promotion of the Quality of Life, San Raffaele Roma Open University, 00166 Rome, Italy; chiara.focaccetti@gmail.com; 3Department of Clinical Sciences and Translational Medicine, University of Rome “Tor Vergata”, 00133 Rome, Italy; monicab4@hotmail.it (M.B.); manzari@med.uniroma2.it (V.M.); bei@med.uniroma2.it (R.B.); 4IRCCS Istituto Giannina Gaslini, 16147 Genoa, Italy; annalisapezzolo56@gmail.com; 5U.O.C. Genetica Medica, IRCCS Giannina Gaslini, 16147 Genoa, Italy; marziaognibene@gaslini.org; 6Department of Paediatric Haematology/Oncology and of Cell and Gene Therapy, Ospedale Pediatrico Bambino Gesù, IRCCS, 00165 Rome, Italy; valentina.folgiero@opbg.net (V.F.); doriana.fruci@opbg.net (D.F.); 7Research Unit of Clinical Immunology and Vaccinology, DPUO, Ospedale Pediatrico Bambino Gesù, IRCCS, 00165 Rome, Italy; nicola.cotugno@opbg.net (N.C.); paolo.palma@opbg.net (P.P.); 8Saint Camillus International University of Health and Medical Sciences, 00131 Rome, Italy

**Keywords:** neuroblastoma, NK cell-mediated immunotherapy, Fas, PVR

## Abstract

**Simple Summary:**

Neuroblastoma (NB) cells adopt several molecular strategies to evade the Natural Killer (NK)-mediated response. Herein, we found that the overexpression of the NF-kB p65 subunit in NB cell lines upregulates the expression of both the death receptor FAS and the activating ligand PVR, thus rendering NB cells more susceptible to NK-cell-mediated apoptosis, recognition, and killing. These data could provide a clue for a novel NK-cell-based immunotherapy of NB. In addition, array CGH analysis performed in our cohort of NB patients showed that loss of both the *FAS* and *PVR* genes correlated with low survival, thus revealing a novel biomarker predicting the outcome of NB patients.

**Abstract:**

High-risk neuroblastoma (NB) is a rare childhood cancer whose aggressiveness is due to a variety of chromosomal genetic aberrations, including those conferring immune evasion. Indeed, NB cells adopt several molecular strategies to evade recognition by the immune system, including the downregulation of ligands for NK-cell-activating receptors. To date, while molecular strategies aimed at enhancing the expression of ligands for NKG2D- and DNAM-1-activating receptors have been explored, no evidence has been reported on the immunomodulatory mechanisms acting on the expression of death receptors such as Fas in NB cells. Here, we demonstrated that transient overexpression of the NF-kB p65 subunit upregulates the surface expression of Fas and PVR, the ligand of DNAM-1, thus making NB cell lines significantly more susceptible to NK-cell-mediated apoptosis, recognition, and killing. In contrast, IFNγ and TNFα treatment, although it induced the upregulation of FAS in NB cells and consequently enhanced NK-cell-mediated apoptosis, triggered immune evasion processes, including the strong upregulation of MHC class I and IDO1, both of which are involved in mechanisms leading to the impairment of a proper NK-cell-mediated killing of NB. In addition, high-resolution array CGH analysis performed in our cohort of NB patients revealed that the loss of *FAS* and/or *PVR* genes correlated with low survival independently of the disease stage. Our data identify the status of the *FAS* and *PVR* genes as prognostic biomarkers of NB that may predict the efficacy of NK-cell-based immunotherapy of NB. Overall, restoration of surface expression of Fas and PVR, through transient upregulation of NF-kB, may be a clue to a novel NK-cell-based immunotherapy of NB.

## 1. Introduction

To efficiently eradicate neuroblastoma (NB), the most common extracranial solid tumor occurring in childhood [1], many efforts are still required. Natural Killer (NK) cells play a crucial role in tumor control since, without prior sensitization, they are able to kill cancer cells [2]. The peculiar cytotoxic abilities of NK cells have enabled the development of several NK-cell-based immunotherapies of cancer [3], including those for NB [4]. Although great successes have been reported, in order to improve the efficacy of NK cells against NB cells, many limitations still need to be overcome.

NK cells mediate the anti-tumor cytotoxicity through two main mechanisms: (i) on the one hand, the recognition of ligands expressed on cancer cells by NK-cell-activating receptors, such as NKG2D and DNAM-1, which triggers lytic granule production; and (ii) on the other hand, the recognition of death receptors expressed on cancer cells by NK cells’ apoptotic ligands, which triggers the apoptotic signaling [5]. In high-risk NB cells, such as those displaying the *MYCN* amplification that is associated with poor prognosis of NB patients [6], ligands for NK-cell-activating receptors are downregulated [7,8], thus conferring on NB cells the ability to evade the NK-cell-mediated innate immune response. In the search for molecular strategies leading to increased expression of ligands for NK-cell-activating receptors in NB cells, the restoration of p53 function led to the upregulation of the ligands ULBP1 and ULBP2 for NKG2D [9] and, as we previously reported, of the ligand PVR (CD155) for DNAM-1 [10]. Of note, PVR is a type I transmembrane glycoprotein belonging to the nectin and nectin-like family of immunoglobulin-like molecules that is expressed in several solid tumors [11] whose recognition by DNAM-1 [12] contributes to the triggering of the NK-cell-mediated release of lytic granules [13]. In contrast, neither the drugs currently used in NB treatment [14] nor JQ1 [15], which aims at reducing the expression level of MYCN, have demonstrated immunomodulatory functions in terms of induction of ligands for NK-cell-activating receptors. Hence, in order to strongly enhance the susceptibility of NB to NK cells, other immunomodulatory mechanisms should be explored.

Fas (also termed CD95, APO-1, and tumor necrosis factor receptor superfamily member 6, or TNFRSF6) is a death receptor expressed on the surface of a variety of nonmalignant tissues and has been implicated in the control of tumor progression [16]. Fas expression is revealed in many tumor cells [17,18], including NB cells [19]. Binding of Fas to FasL, expressed on lymphocytes [20] including NK cells [21,22], triggers the receptor trimerization leading to the formation of the death-inducing signaling complex and, subsequently, by inducing the apoptosis of malignant cells [20,23,24]. The polymorphisms revealed in the gene promoter of both Fas and FasL have been correlated with increased development of hematopoietic and solid cancers, including promyelocytic leukemia, acute myeloid leukemia, head and neck squamous cell carcinoma, and pancreatic cancer [25,26,27,28,29]. Furthermore, loss of Fas function, known to cause autoimmune lymphoproliferative syndrome [30], contributes to tumor immune escape and correlates with poor prognosis [31,32,33]. Thus, restoration of Fas function, in advanced tumors in which loss of Fas function occurs, should represent a novel strategy to enhance NK-cell-mediated NB eradication. Of note, NK cells play a crucial role in the cancer’s eradication and metastasis control through FasL-mediated cell death [34,35].

Fas expression is directly regulated by the nuclear factor kappa B (NF-kB) [36,37], which, although known as a tumor-promoting transcription factor [38], has also been widely described as an inducer of senescence, apoptosis, and cell death [39,40,41,42,43,44,45]. The p65 subunit (Rel A), a component of the NF-kB heterodimer, has a crucial role in inducing the Fas expression, as demonstrated in a colon carcinoma cell model [37]. Furthermore, NF-kB has shown further immunomodulatory functions as it acts as a direct transcription factor of PVR, as evaluated in multiple myeloma cells [46]. Herein, we overexpressed the NF-kB p65 subunit in NB cells and evaluated the surface expression levels of Fas along with those of ligands for NKG2D- and DNAM-1-activating receptors, including PVR, and, thus, the susceptibility of NB cells to NK-cell-mediated killing. In addition, we evaluated the effect of two cytokines IFNγ and TNFα, known to upregulate Fas [47,48,49,50,51,52] through the activation of NF-Kb [53,54], on both the surface expression of activating ligands and FAS and on NK-cell-mediated apoptosis and recognition of NB cell lines. Furthermore, we investigated the correlation of *FAS* and *PVR* gene status with the disease risk and survival of NB patients by a high-resolution array CGH analysis. Our data suggest that the induction of both Fas and PVR by the NF-kB p65 subunit might represent a novel therapeutic strategy to enhance the NK-cell-mediated killing of NB cells.

## 2. Materials and Methods

### 2.1. NB Cell Lines, NK Cells, and Reagents

The following human NB cell lines were used in this study: GICAN, ACN (Interlab Cell Line Collection, Banca Biologica, and Cell Factory (www.iclc.it, accessed on 01 February 2020)), SK-N-AS, SH-SY5Y, SH-EP, SK-N-SH, SK-N-BE(2)c, IMR-32 (American Type Culture Collection (ATCC)), LA-N-1 (Creative Bioarray), LA-N-5 (the Leibniz-Institut DMSZ), and SMS-KCNR (Children’s Oncology Group Cell Culture). All NB cell lines were characterized by: (i) HLA class I typing by PCR-SSP sets (Genovision As, Oslo, Norway) according to the instructions of the manufacturer; and (ii) array CGH. The K562 cell line (ATCC) was used as a control target for NK cell functional assays. Cell lines were cultured in RPMI 1640 medium supplemented with 10% FBS (Thermo Fisher Scientific, Waltham, MA, USA), 2 mM glutamine, 100 mg/mL penicillin, and 50 mg/mL streptomycin (Euro Clone S.p.A., Milan, Italy). 

Human NK cells were isolated from peripheral blood of healthy donors by using the RosetteSep NK-cell enrichment mixture method Kit (StemCell Technologies, Vancouver, BC, Canada) and Ficoll-Paque Plus (Lympholyte Cedarlane, Burlington, ON, Canada) centrifugation, then checked for CD3^−^CD56^+^ immunophenotype and the expression of the activating receptors NKG2D and DNAM-1 and the inhibitory receptors NKG2A, KIR2DL1, KIR2DL3, and KIR3DL1 by flow cytometry. NK cells with greater than 90% purity and positive for all four inhibitory receptors were resuspended in NK MACS medium (Miltenyi Biotec, Bergisch Gladbach, Germany) supplemented with NK MACS supplement, AB serum, and 500 IU/mL of recombinant human IL-2 (PeproTech, Rochy Hill, NJ, USA). NK cells were cultured at 37 °C, divided every three days, and used up to 20 days after isolation for experiments. All NK cell function assays were performed in the alloreactivity setting [55,56].

### 2.2. Antibodies, Cytokines, Western Blotting, Apoptosis, and Flow Cytometry

The following antibodies were used: anti-p65 (NB100-2176) purchased from Novus Biologicals, anti-Actin (I-19) purchased from Santa Cruz Biotechnology, anti-IDO1 (D5J4E) from Cell Signaling, and anti-tubulin (TU-02) from Immunological Sciences for Western blotting; anti-CD107a-APC (H4A3), anti-CD3-Alexa Fluor-700 (UCHT1), anti-CD56-PE-Cy7 (B159), anti-CD45 (HI30), and anti-NKG2D-BV605 (1D11) purchased from BD Biosciences, anti-DNAM-1-APC (11A8) purchased from Biologend, anti-KIR2DL1/2DS1-PC5.5 (EB6B) and anti-KIR2DL2/L3/S2-PE (GL-183) purchased from Beckman Coulter (Brea, CA, USA), anti-NKG2A-Alexa Fluor-700 (131411), anti-KIR3DL1-APC (DX9), anti-ULBP1-PE (170818), anti-ULBP2/5/6-PE (165903), anti-ULBP3-PE (166510), anti-MICA (159227), anti-MICB (236511), anti-TRAIL/R2-APC (17908), anti-CD155/PVR-PE (300907), anti-Nectin-2/CD112-APC (610603), and anti-FAS (DX2) purchased from R&D Systems, and goat F(ab’)2 Fragment anti-mouse IgG FITC (IM1619) purchased from Dako for flow cytometry. For some experiments, NB cell lines were treated for 48 h with 50 ng/mL IFNγ (285-IF, R&D Systems) and 20 ng/mL TNFα (210-TA, R&D Systems). The apoptosis state of tumor cells was evaluated by using a PE-conjugated Annexin V apoptosis detection Kit (Biolegend, San Diego, CA, USA) and analyzed by flow cytometry.

For Western blotting experiments, whole-cell extracts were quantified by bicinchoninic acid (BCA) assay (Thermo Fisher Scientific), resolved by 8–10% sodium dodecyl sulfate polyacrylamide gel electrophoresis (SDS-PAGE), electroblotted, and then filters probed with primary antibodies followed by HRP-conjugated goat anti-mouse IgG (Jackson) [57]. Flow cytometry was performed by using FACSCantoII, LSRFortessa (BD Bioscences, San Jose, CA, USA), and Cytoflex (Beckman Coulter) and analyzed by FlowJo Software (BD, Ashland, OR, USA).

### 2.3. Patient Samples and Genomic Profile Analysis

The present study was conducted in line with the Declaration of Helsinki. For each NB patient, written informed consent provided by parents and Ethical approval by the Institutional Committee were obtained. Demographic, molecular, and histologic features of the cases studies are detailed in Table 2. Staging and histological classification were performed according to the International Neuroblastoma Staging System (INSS) and the International Neuroblastoma Pathology Classification (INPC, [58,59]), respectively. NB samples were stored in the BIT-neuroblastoma Biobank of IRCCS Giannina Gaslini Genoa and evaluated before treatment at the time of diagnosis. Tumor content was confirmed by a local pathologist by evaluation of tumor sections stained with hematoxylin and eosin. Tumor DNA was isolated from fresh NB tissue using the MasterPure DNA Purification Kit (Epicentre-Illumina, San Diego, CA, USA) according to the manufacturer’s instructions.

DNAs from NB primary tumors were evaluated by high-resolution array comparative genomic hybridization (a-CGH), in accordance with current guidelines [60], to measure *MYCN* amplification together with *FAS*, *PVR,* and *RELA* gene status. For these evaluations, we used a 4 × 180K platform (Agilent Technologies, Santa Clara, CA, USA) with the mean resolution of approximately 25 kb. Based on mapping positions by reference to the Genome Assembly GRCh38/hg19 (UCSC Genome Browser, http://genome.ucsc.edu, accessed on 1st February 2020), a copy number variant was defined as a shift of at least 8 consecutive probes. The Genomic Workbench 7.0.40 software (Agilent, Santa Clara, CA, USA) was used to analyze data. ADM-1 (threshold 10) was used to assess altered chromosomal regions and breakpoint events, and a 0.5 Mb window was considered to reduce false positives. Test quality was assessed by the strength of QCmetrics values. Polymorphisms (http://projects.trag.ca/variation/, accessed on 1st February 2020) were not included in this study because they were included in normal variants.

### 2.4. Evaluation of Quantitative mRNA Expression 

TRIzol reagent (Thermo Fisher Scientific) was used to extract total RNA. A SuperScript II First Strand cDNA Synthesis Kit (Thermo Fisher Scientific) was used to synthesize the first cDNA strand. TaqMan gene expression assays (Applied Biosystems) were used to perform Quantitative real-time PCR (qPCR) reactions (Thermo Fisher Scientific, (Hs00197846_m1 for PVR, Hs00236330_m1 for FAS)). The 2^−∆∆Ct^ method and 2^−∆Ct^ as the expression level were used to determine the relative gene expression considering GAPDH (Hs02758991_g1) as an endogenous control.

### 2.5. Plasmids and Transfection

NF-kB p65 vector, kindly provided by Dr Massimo Levrero (Rome Oncogenomic Center (R.O.C.), Regina Elena Institute, Rome, Italy), and the corresponding empty vector (pcDNA3) were used to transfect NB cell lines using Lipofectamine 2000 (Invitrogen, Waltham, MA, USA). NB cells, cultured on tissue culture plates, were transfected 24 h later at an 80% confluence by using DNA–lipofectamine complexes diluted in OptiMEM (Thermofisher). At fifteen hours after transfection, the culture medium was replaced with fresh medium. NB p65-transfected cells were then evaluated for the apoptosis state, for the expression of ligands recognized by NK-cell-activating receptors and used for NK cell function experiments.

### 2.6. Apoptosis, Degranulation, and Cytotoxicity Assay

NK cells isolated from healthy donors, expanded and activated in vitro, were used for functional assays by co-culturing them with NB cells and, then, by evaluating the NK-cell-mediated degranulation in response to target cells and both the apoptosis and cytotoxicity of NB cells.

The apoptotic state of tumor cells was evaluated by co-culturing NK cells with target cells in a 1:1 ratio for 4 h. Cells were stained with anti-CD45, Annexin V, and 7-ADD and analyzed by flow cytometry. The apoptotic state of NB cells was evaluated in the CD45^−^ subset. 

To perform the degranulation assay, NK cells were co-cultured with target cells at a 1:1 ratio for 3 h in complete medium in the presence of anti-CD107a at a 1:100 dilution. During the last 2 h of co-culture, GolgiStop (BD Bioscence), used at a 1:500 dilution, was added. Cells were then washed, centrifuged, and stained with anti-CD56 and anti-CD45 to evaluate CD107a expression in the CD56^+^CD45^+^ subset by flow cytometry. For the blocking experiments, NK cells were pretreated for 20 min with 25 μg/mL of neutralizing anti-DNAM-1 (DX11, BD-Parmingen, Franklin Lakes, NJ, USA) or anti-FasL (NOK-1, Biolegend) before co-culture with target cells.

The cytotoxic activity of NK cells was tested with a standard 4-h ^51^Cr-release assay. Target cells (5 × 10^3^) labeled with ^51^Cr (Amersham International) were co-cultured in 96-well plates (round bottom) with NK cells at different effector cell-to-target (E:T) ratios and, then, incubated at 37 °C. At 4 h after co-culture, plates were gently centrifuged to transfer 25 μL of supernatant/well in LumaPlate-96 (PerkinElmer Life Sciences, Boston, MA, USA) and finally to measure the ^51^Cr release by a TopCount NXT beta detector (PerkinElmer). Each experimental group was analyzed in triplicate, and the percentage of specific lysis was evaluated by counts per minute (cpm) and determined as follows: 100 × (mean cpm experimental release−mean cpm spontaneous release)/(mean cpm total release−mean cpm spontaneous release). The specific lysis was converted into lytic units (L.U.), extrapolated from the curve of the lysis percentage. A lytic unit corresponds to the number of NK cells required to lyse 20% of 10^6^ target cells during 4 h of incubation.

### 2.7. Statistical Analysis

For all data, statistical significance was evaluated with the unpaired and paired two-tailed Student’s t-test. Normalized values were analyzed for correlation by the regression analysis using GraphPad software. The association of the *FAS* and/or *PVR* gene status and the patient’s outcome was evaluated by Fisher’s exact test. *p* values not greater than 0.05 were considered to be statistically significant.

## 3. Results

### 3.1. NF-kB p65 Subunit Enhances the Expression of FAS and PVR in NB Cell Lines

To assess the correlation between Fas, PVR, and the NF-kB p65 subunit, the status of genes encoding Fas (*FAS*, chromosomal coordinates chr10: 88,968,429–89,017,059; cytoband 10q23.31), PVR (*PVR*, chromosomal coordinates chr19: 44,643,798–44,666,162; cytoband 19q13.31), and p65 (*RELA,* chromosomal coordinates chr11: 65,653,597–65,663,090; cytoband 11q13.1) was evaluated in a panel of 11 NB cell lines, including *MYCN* non-amplified (SH-SY-5Y, ACN, SK-N-AS, and SH-EP), *MYCN* gain (SK-N-SH and GICAN), and *MYCN*-amplified (LA-N-5, SMS-KCNR, IMR-32, SK-N-BE(2)c, and LA-N-1). High-resolution array CGH (a-CGH) analysis revealed that the *FAS* gene was lost in SK-N-AS, SK-N-BE(2)c, and LA-N-1, it was gained in SH-EP and IMR-32, and it was present in a single copy in the remaining NB cell lines (Table 1).

The *PVR* gene was lost in GICAN, SK-N-BE(2)c, and LA-N-1, it was gained in ACN, and it was present as a single copy in the remaining NB cell lines (Table 1). The *RELA* gene was present in a single copy in all NB cell lines with the exception of GICAN where it was gained and in SK-N-BE(2)c and LA-N-1 where it was lost (Table 1). Interestingly, in these latter two NB cell lines, which are both *MYCN*-amplified and thus belong to the NB cell type most resistant to anticancer treatments [61], the *FAS, PVR,* and *RELA* genes were lost (Table 1). In line with the *RELA* gene status, Western blotting analysis revealed lower p65 expression levels in SK-N-BE(2)c and LA-N-1 (p65/actin expression ratio: 0.2 and 0.5, respectively) and higher expression levels in GICAN (p65/actin expression ratio: 1.8) compared with other NB cells (Appendix A).

We asked whether NF-kB-induced Fas expression could affect the susceptibility of NB cells to NK cells. For this purpose, four NB cell lines, two *MYCN* non-amplified as SH-SY-5Y and SK-N-AS, and four *MYCN* amplified as LA-N-5, SMS-KCNR, IMR-32, and SK-N-BE(2)c, were transfected with pcDNA3 carrying the gene encoding for the p65 subunit or pcDNA3 as a control plasmid (a representative dose/response experiment of Western blotting analysis after p65 gene transfection into SH-SY-5Y and LA-N-5 is shown in Appendix A) and evaluated for the expression of Fas and ligands for NK-cell-activating receptors, such as MICA, MICB, ULBP1, ULBP2/5/6, and ULBP3 for NKG2D and PVR and Nectin-2 for DNAM-1 (Figure 1A,B and Appendix A). The efficacy of transfection with the plasmid pcDNA3 carrying the p65 gene was monitored by evaluating the increased surface expression of the major histocompatibility complex (MHC) class I, which is highly sensitive to NF-kB levels in NB, as we previously reported [62,63]. TRAIL-R2 expression was evaluated as an additional apoptosis-inducing receptor [64]. Transfection of the p65 gene significantly upregulated the expression of Fas and PVR in all NB cells (Figure 1A,B), with the exception of Fas in SK-N-AS, and of both Fas and PVR in the SK-N-BE(2)c cell line, in agreement with the loss of *FAS* and *PVR* genes as revealed by a-CGH analysis (Table 1). The upregulation of Fas and PVR was more evident in *MYCN* non-amplified NB cell lines (with the exception of Fas in SK-N-AS) than in those *MYCN*-amplified, as also confirmed by RT-PCR (Appendix A), thus suggesting that the *MYCN* amplification might affect the p65-mediated induction of both Fas and PVR. Conversely, the expression of ligands for both NKG2D- and DNAM-1-activating receptors and for TRAIL-R2 was unaltered with the exception, to a lesser extent, of ULBP1 in the SH-SY-5Y and SK-N-AS cell lines, ULBP3 in the SK-N-AS cell line, and TRAIL-R2 in the SH-SY-5Y cell line (Figure 1B). Collectively, these data indicate that the transient overexpression of the NF-kB p65 subunit induced an immunomodulatory effect by upregulating both Fas and PVR in NB cell lines carrying intact *FAS* and *PVR* genes.

### 3.2. NF-kB p65 Subunit Overexpression Renders NB Cell Lines More Susceptible to NK-Cell-Mediated Recognition and Killing

To evaluate whether upregulation of Fas and PVR mediated by the transient increased expression of the NF-kB p65 subunit could affect the susceptibility of high-risk NB cell lines to NK-cell-mediated recognition and killing, *MYCN* non-amplified SH-SY-5Y and ACN cell lines and *MYCN*-amplified LA-N-5 and SMS-KCNR NB cells were transfected with the gene encoding for p65 and used as targets in NK-cell-mediated apoptosis, degranulation, and cytotoxicity assays. Furthermore, SK-N-BE(2)c cells, being lost for *FAS* and *PVR* genes (Table 1) and whose expression was therefore unaffected by p65 gene transfection (Figure 1A), were used as a negative control. Both *MYCN* non-amplified and *MYCN*-amplified cells overexpressing p65 were significantly more susceptible to NK-cell-mediated apoptosis (Figure 2A,B), recognition (Figure 2C,D), and lysis (Figure 2E,F) than pcDNA3-transfected control cells. In contrast, p65-overexpressing SK-N-BE(2)c cells were recognized similarly to the pcDNA3-SK-N-BE(2)c control cells, thus suggesting that the upregulation of Fas and PVR, through the overexpression of the p65 subunit, specifically increased the susceptibility of *MYCN*-amplified NB cell lines to NK-cell-mediated apoptosis, recognition, and killing. Furthermore, blocking experiments demonstrated that both DNAM-1 and FasL were involved in NK-cell-mediated apoptosis and recognition of p65-overexpressing NB cell lines (Figure 2A–D).

### 3.3. IFNγ and TNFα Treatment of NB Cell Lines Increases Fas and Renders Them More Susceptible to NK-Cell-Mediated Apoptosis but Not to NK Cell Degranulation

In the search for clinical approaches that lead to the upregulation of Fas and PVR, thereby rendering NB cells more susceptible to NK-cell-mediated killing, we tested the efficacy of the two cytokines IFNγ and TNFα, known to upregulate Fas [47,48,49,50,51,52] in different contexts through the activation of NF-Kb [53,54]. The treatment of *MYCN* non-amplified SH-SY-5Y, ACN, SK-N-AS, SHE-P, and SK-N-SH and *MYCN-*amplified LA-N-5, IMR-32, and SK-N-BE(2)c NB cell lines with IFNγ and TNFα for 48 h induced a modulation of the p65 expression level with an increase in the ACN, SK-N-AS, SHE-P, and, to a lesser extent, LA-N-5 cell lines (Figure 3A). Interestingly, the IFNγ and TNFα treatment significantly increased the expression of Fas in all NB cell lines with the exclusion of SK-N-BE(2)c used as a negative control, but of PVR only in the SK-N-AS cell line (Figure 3B).

In addition, the cytokine treatment induced an increase in ULBP1 in SH-SY-5Y, SK-N-AS, and IMR-32, in ULBP3 in LA-N-5, and in Nectin-2 in IMR-32 (Appendix A). By contrast, the cytokine treatment induced the downregulation of ULBP2/5/6 in SH-SY-5Y and LA-N-5, of ULBP-3 in SH-SY-5Y, of Nectin-2 in SH-SY-5Y, ACN, and LA-N-5, and of TRAIL-R2 in all NB cell lines with the exception of SK-N-BE(2)c (Appendix A). As expected, MHC class I expression was strongly upregulated by the cytokine treatment in all NB cell lines (Figure 3B), as previously reported [62,65]. Accordingly, all NB cell lines treated with IFNγ and TNFα (with the exclusion of SK-N-BE(2)c) were significantly more susceptible to NK-cell-mediated apoptosis than untreated cells (Figure 3C), whereas the cytokine treatment alone did not affect the apoptotic status of NB cell lines (Appendix A). In contrast, the NK-cell-mediated degranulation in response to IFNγ- and TNFα-treated NB cells was significantly reduced, except for SK-N-AS and SK-N-BE(2)c in which it was unchanged. These data suggest that the downregulation of ligands for NK-cell-activating receptors (Appendix A) as well as the strong upregulation of MHC class I upon cytokine treatment (Figure 3B) contributed to the impaired NK cell degranulation in response to cytokine-treated NB cell lines.

Next, to explore further mechanisms of immune evasion leading to the impaired NK cell degranulation in response to cytokine-treated NB cells, we assessed the expression level of Indoleamine-pyrrole 2,3-dioxygenase1 (IDO1), an enzyme involved in mechanisms conferring resistance to immune cell activities in most tumors [66], including NB as we previously reported [67]. All NB cell lines treated with IFNγ and TNFα showed a strong upregulation of IDO1 (Figure 3E), indicating that the treatment with these cytokines, although it resulted in an increased Fas expression and consequently enhanced NK-cell-mediated apoptosis of NB cell lines, controlled the NK-cell-mediated recognition process through IDO1-dependent immune evasion mechanisms, thereby attenuating the NK-cell-mediated killing. These data suggest that, to induce Fas and activate ligands such as PVR on NB cells, translational approaches based on the use of cytokines should be reviewed and the combined use of IDO1 inhibitors [68] should optimize the NK-cell-mediated killing of cytokine-treated NB cells. Overall, these data indicate that many more investigations in the search for drugs that can induce Fas and PVR by NF-Kb induction are warranted.

### 3.4. Loss of FAS and PVR Genes Correlates with Poor Prognosis and Low Survival

Next, we analyzed a cohort of 280 NB primary tumors carrying recurrent structural chromosomal aberrations (SCAs) for the loss of chromosome 10q containing the *FAS* gene and the loss of chromosome 19q containing the *PVR* gene at disease onset by a-CGH, derived from the analysis of 465 NB primary tumors. The remaining 185 primary NB tumors showed numerical chromosome aberrations (NCAs). Patients carrying SCAs were mostly in Stage 4 (130 patients) and in Stage 3 (67 patients), as assessed by the International Neuroblastoma Staging System (INSS) parameters [58,59]. Of the patients carrying SCAs, 32 showed loss of chromosome 10q and/or loss of chromosome 19q and 69% of these (22 patients) were devoid of the *FAS* and/or *PVR* genes (Table 2). Specifically, one patient showed loss of both *FAS* and *PVR* genes, 12 showed loss of only the *FAS* gene, including one with copy-neutral loss of heterozygosity (cnLOH), and 9 patients showed loss of only the *PVR* gene (Table 2). Interestingly, loss of one or both genes was observed in most NB patients who died (17 out 22). This outcome occurred in the only patient with loss of both *FAS* and *PVR* genes, in 67% of patients (8 out of 12) with loss of the *FAS* gene, and in 89% of patients (eight out of nine) with loss of the *PVR* gene (Figure 4A, right panel).

Only two of the five alive patients, displaying *FAS* or *PVR* gene deletions, were in complete remission (CR), while the remaining three showed an active disease (AD) (Table 2). Conversely, *RELA* gene status was normal in most of these patients, with the exception of patients 3, 15, 22, and 27 in whom it was gained (Appendix A). Most of the patients with a fatal outcome had Stage 4 or Stage 3 (13 out of 22). They included the patient with loss of both genes, six patients with loss of *FAS,* and six patients with loss of *PVR*. Interestingly, four patients with Stage 1 and 2 and loss of either the *FAS* or the *PVR* gene also had a fatal outcome after relapse (Figure 4B). Conversely, patients carrying the loss of chromosome 10q or chromosome 19q, but intact *FAS* or *PVR* genes (10 out of 32), which include also those with Stage 4, had CR (Figure 4A, left panel). Interestingly, in this group of patients, one was at Stage 4s, a tumor characterized by a high rate of spontaneous regression [69] (Table 2).

Statistical analysis comparing the number of patients in early stages (Stage 1 and Stage 2) and those in advanced stages (Stage 3 and Stage 4) revealed that loss of *FAS* and/or *PVR* genes was not associated with NB patient stage (Appendix A, Figure 4B). Furthermore, while 77.3% (17 out of 22 patients) of cases with loss of *FAS* and/or *PVR* genes had a fatal outcome, only 18.9% (49 out of 258 patients) of SCA NB patients, carrying intact *FAS* and/or *PVR* genes, died at 5 years of follow-up (Table 3, Figure 4C). Thus, loss of *FAS* and/or *PVR* genes was associated with poor outcome (*p* < 0.00001).

Overall, these data indicate that the loss of *FAS* and *PVR* genes occurs independently of disease stage and correlates with low survival in NB patients.

## 4. Discussion

NB is one of the most aggressive forms of cancer arising in childhood, and still accounts for 15% of all pediatric cancer deaths with a 3-year event-free survival loss of more than 40%. Although many clinical efforts are currently underway, including surgery, radiation therapy, chemotherapy, and stem cell transplantation, relapse is very common, and the outcome of high-risk disease remains poor. NK cells play a crucial role in fighting NB as evaluated in several preclinical and clinical studies [4,70,71,72].

The aggressiveness of NB is due to many molecular mechanisms. In addition to chromosomal aberrations such as the *MYCN* amplification conferring proliferative and metastatic abilities to tumor cells [73], NB cells adopt several molecular strategies to evade the immune control, including the downregulation of ligands for NK-cell-activating receptors, thus preventing proper NK-cell-mediated recognition and killing of NB cells [8,74]. We previously reported that Nutlin-3a, a nontoxic small molecule that antagonizes the inhibitory interaction of MDM2 with the tumor suppressor p53, induces the expression of ligands for NK-cell-activating receptors, thus rendering NB cells more susceptible to NK-cell-mediated killing [10]. Furthermore, p53 is a direct transcription factor for PVR [10], a ligand recognized by DNAM-1-activating receptors, the binding of which is crucially involved in signaling leading to the production of lytic granules, thus allowing NK cells to mediate the killing of tumor cells [13]. In contrast, neither the current drugs used in the clinical treatment of NB [14] nor the BET-bromodomain inhibitor JQ1 [15] have shown immunomodulatory proprieties leading to an increase in NB susceptibility to NK cells. In the search for molecular strategies to enhance the expression of molecules involved in the NB recognition by NK cells, thereby promoting the NK-cell-mediated eradication of NB, we focused on Fas, a death receptor crucially involved in the NK-cell-mediated apoptosis of tumor cells [21,22]. Of note, both the expression and function of Fas are frequently impaired in cancer cells [25,26,27,28,29,31,32,33], further contributing to the evasion from the innate immune response. This suggests that restoration of Fas expression might represent a molecular strategy leading to improved susceptibility of tumor cells to NK-cell-mediated immune control.

Herein, we provided evidence that the transfection of NB cells with the p65 subunit, a component of the NF-kB heterodimer that is known to directly regulate the expression of both Fas [36,37] and PVR [46], enhanced the NB susceptibility to NK-cell-mediated killing through DNAM-1 and FasL engagement as demonstrated by blocking experiments. Overexpression of the p65 subunit led to a significant increase in the surface expression of Fas and PVR in *MYCN*-amplified NB cell lines. Of note, loss of both *FAS* and *PVR* genes was detected in high-risk NB cell lines, one of which (SK-N-BE(2)c) was not affected by p65 transfection, either in terms of Fas and PVR expression or susceptibility to NK-cell-mediated killing. Furthermore, the treatment of NB cell lines with the cytokines IFNγ and TNFα, known to upregulate Fas expression [47,48,49,50,51,52] through NF-Kb induction [53,54], enhanced the NK-cell-mediated apoptosis but not the NK-cell-mediated degranulation. The impaired NK-cell-mediated degranulation was due to a dual mechanism of innate immune evasion: (i) the strong upregulation of MHC class I, which is known to cause an imbalance of activating versus inhibitory signals on NK cells [75] accompanied by downregulation of activating ligands at different levels; and (ii) the strong upregulation of IDO1, a molecule involved in most of the immune evasion mechanisms of tumors [66], including NB [67]. Thus, in the clinical setting, drugs that can control IDO1 activity [68] should potentiate the Fas-dependent NK-cell-mediated killing of NB cells treated with cytokines such as IFNγ and TNFα.

The loss of both *FAS* and *PVR* genes was evaluated in a cohort of NB patients carrying SCAs and the loss of 10q and 19q chromosomes and was significantly associated with low survival. Interestingly, among these patients, not only those with an advanced disease stage (Stage 4 and 3), but also patients with lower disease risk (Stage 2 and 1), relapsed and died. In contrast, patients who carried only the loss of chromosome 10q or chromosome 19q but retained both *FAS* and *PVR* genes showed a better outcome and complete remission. This evidence reveals that the status of *FAS* and *PVR* genes may represent a novel biomarker predicting the outcome of NB patients, thus suggesting that their restoration could improve disease status through an immunomodulatory action that could involve NK-cell-mediated innate anti-tumor functions.

All these data indicate that NF-kB upregulation may promote NK-cell-mediated recognition and killing of NB cells through increased surface expression of both Fas and PVR. This suggests that two death signals are upregulated in this p65-mediated process: (i) one triggered by the apoptosis pathway occurring in NB cells following Fas and FasL binding, due to increased Fas expression; and (ii) the second triggered by the release of lytic granules from NK cells following the PVR and DNAM-1 binding due to increased PVR expression. Since we used cytokine-activated NK cells, these two mechanisms can occur simultaneously, within three hours after contact of activated NK cells with NB cells, as previously reported [76]. This dual signal of activation of NB cell death may be an effective approach to help overcome the various mechanisms of NK-cell-mediated resistance to killing that occur in cancer cells [5,77].

The role of NF-kB in cancer is controversial [78]. Several studies have revealed that NF-kB is involved in several pathways leading to apoptosis, necroptosis, autophagy, and inflammation in the tumor microenvironment that, depending on tissue-specific contexts and differential stages of cancer development, are associated with tumor progression or suppression [79,80,81,82]. Of note, NF-kB restoration is critical to increase the expression of MHC class I [62,63], whose downregulation in tumor cells represents one of the main strategies adopted by cancer to evade the adaptive immune response mediated by CD8^+^ T cells [83]. Accordingly, several molecules aimed at stimulating NF-kB such as TNFα [84], retinoids [85,86], betulinic acid [87], Nedd4 binding protein 1 (N4BP1) [88], and IL-17 [89] are being investigated for novel immunotherapies of solid cancer, including NB [86,88]. Our results support a pro-apoptotic and anti-cancer role of the NF-kB p65 subunit in NB cells, via transient transfection, through enhanced NK-cell-mediated recognition and killing of NB cells following upregulation of Fas and PVR. These data suggest that, under specific therapeutic circumstances that induce NF-kB function, an immunomodulatory effect could be achieved, thereby increasing the susceptibility of NB cells to NK-cell-mediated killing, thus representing a novel approach aimed at supporting the NK-cell-mediated immunotherapy of NB.

## 5. Conclusions

Herein, we provided evidence that the loss of *FAS* and *PVR* genes was linked to a poor outcome and low survival of NB patients. The restoration of surface expression of both Fas and PVR could be mediated by the transient overexpression of the NF-kB p65 subunit in NB cell lines, which made NB cells more susceptible to NK-cell-mediated recognition and killing. Our model suggests that transient induction of NF-kB in NB cells has an immunomodulatory effect through the upregulation of Fas and PVR, thus representing a clue for a novel therapeutic approach aimed at enhancing the NK-cell-mediated eradication of NB.

## Figures and Tables

**Figure 1 cancers-13-04368-f001:**
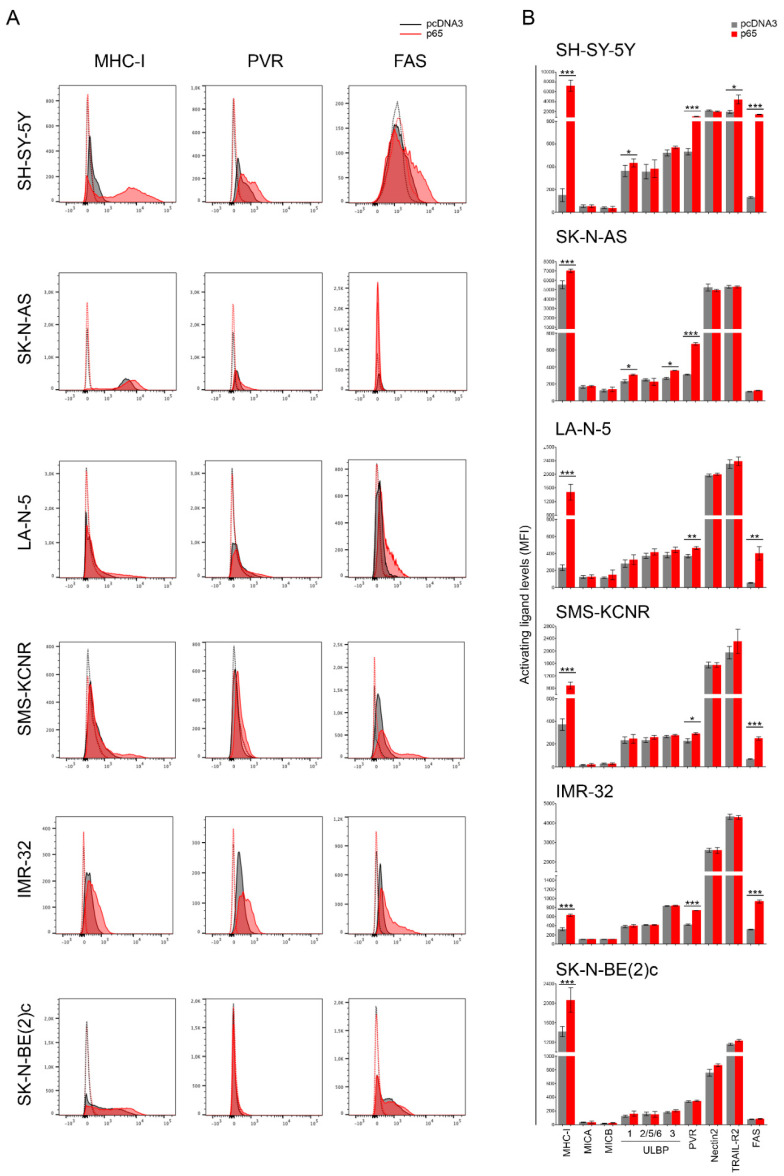
NF-kB p65 subunit enhances the expression of FAS and PVR in NB cell lines. (**A**) NB cell lines SH-SY-5Y, SK-N-AS, LA-N-5, SMS-KCNR, IMR-32, and SK-N-BE(2)c were transiently transfected for 48 h with either the empty vector pcDNA3 or the vector expressing the p65 subunit and then evaluated for the surface expression of MHC class I, the indicated ligands for NK-cell-activating receptors, TRAIL-R2, and Fas by flow cytometry analysis (Appendix A). Isotype-matched negative control antibodies are displayed as dashed gray and red lines for pcDNA3- and p65-transfectetd NB cell lines, respectively. A representative experiment out of six performed is shown. (**B**) Summary of six independent flow cytometry analyses. MFI, mean of fluorescence intensity. Mean + SD; * *p* < 0.05; ** *p* < 0.01; *** *p* < 0.001. *p* value (two-tailed unpaired Student’s *t* test).

**Figure 2 cancers-13-04368-f002:**
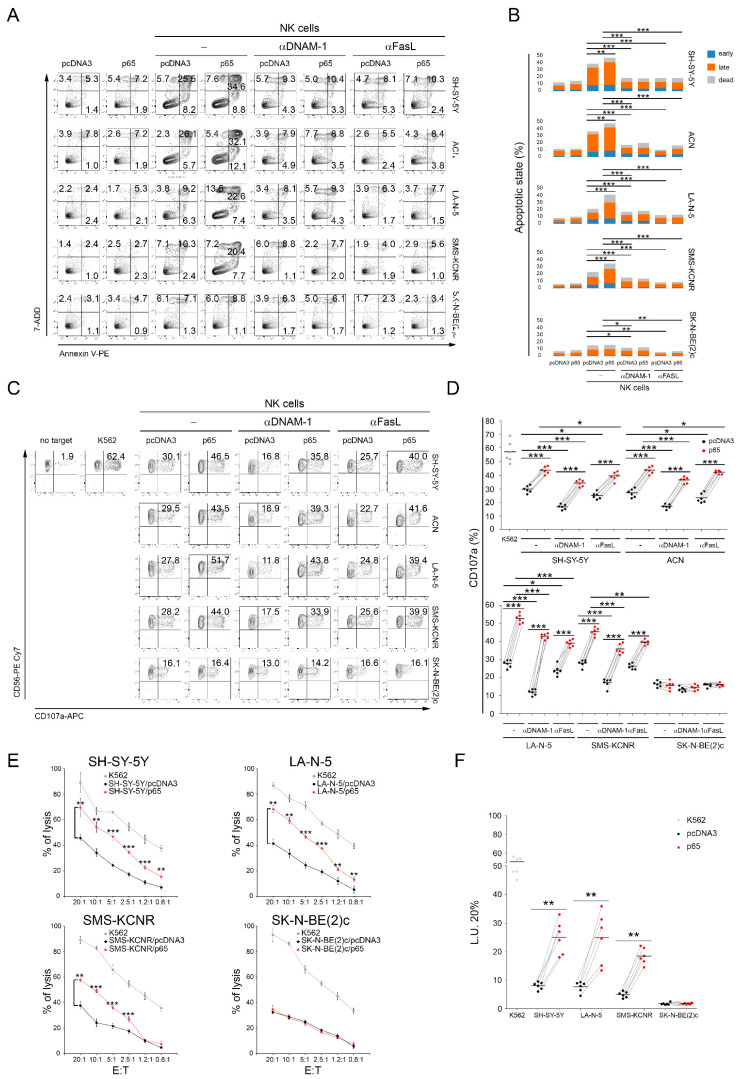
Transfection of the NF-kB p65 subunit renders NB cell lines more susceptible to NK-cell-mediated apoptosis, recognition, and killing. (**A**) NB cell lines, transiently transfected for 48 h with either an pcDNA3 empty vector or the vector expressing the p65 subunit, were co-cultured for 4 h with untreated (−) or anti-DNAM-1 or anti FasL-treated NK cells, and then stained with anti-CD45, Annexin V, and 7-ADD. The apoptotic state of CD45^−^ was evaluated by flow cytometry. A representative experiment, out of six performed, for each tested NB cell line indicated, is reported. The percentage of cells in the early phase (Annexin V^+^ 7-ADD^−^) and the late phase (Annexin V^+^ 7-ADD^+^) of apoptosis as well as dead cells (Annexin V^−^ 7-ADD^+^) is reported in each plot. (**B**) Summary of six independent experiments is reported in the stacked histogram; * *p* < 0.05, ** *p* < 0.01; *** *p* < 0.001. (**C**) Degranulation of human CD45^+^CD56^+^CD3^−^ NK cells, untreated (−) or anti-DNAM-1 or anti FasL-treated, measured as CD107a cell-surface expression upon co-culture with the indicated NB cell lines transiently transfected for 48 h with either an empty pcDNA3 vector or the vector expressing the p65 subunit. The K562 cell line was used as a target for the positive control. The percentage of CD107a^+^ NK cells is indicated in each plot. A representative experiment out of six independently performed is shown. (**D**) Summary of degranulation of NK cells isolated from peripheral blood of six healthy donors. Dots correspond to percentage of CD107a^+^ NK cells from each healthy donor; horizontal bars indicate the mean; * *p* < 0.05, *** *p* < 0.001. (**E**) NB cells transiently transfected for the empty pcDNA3 vector or the vector expressing the p65 subunit were used as target cells for NK cells at the indicated effector:target (E:T) ratios by 4-h ^51^Cr-release assay. A representative experiment out of six independently performed is shown. Mean ± SD; ** *p* < 0.01; *** *p* < 0.001. (**F**) Summary of cytotoxic assay of NK cells isolated from peripheral blood of six healthy donors. Specific lysis, converted to L.U. 20%, is indicated by dots. L.U. 20% of the E:T unpaired test; horizontal bars, mean; ** *p* < 0.01; *** *p* < 0.001.

**Figure 3 cancers-13-04368-f003:**
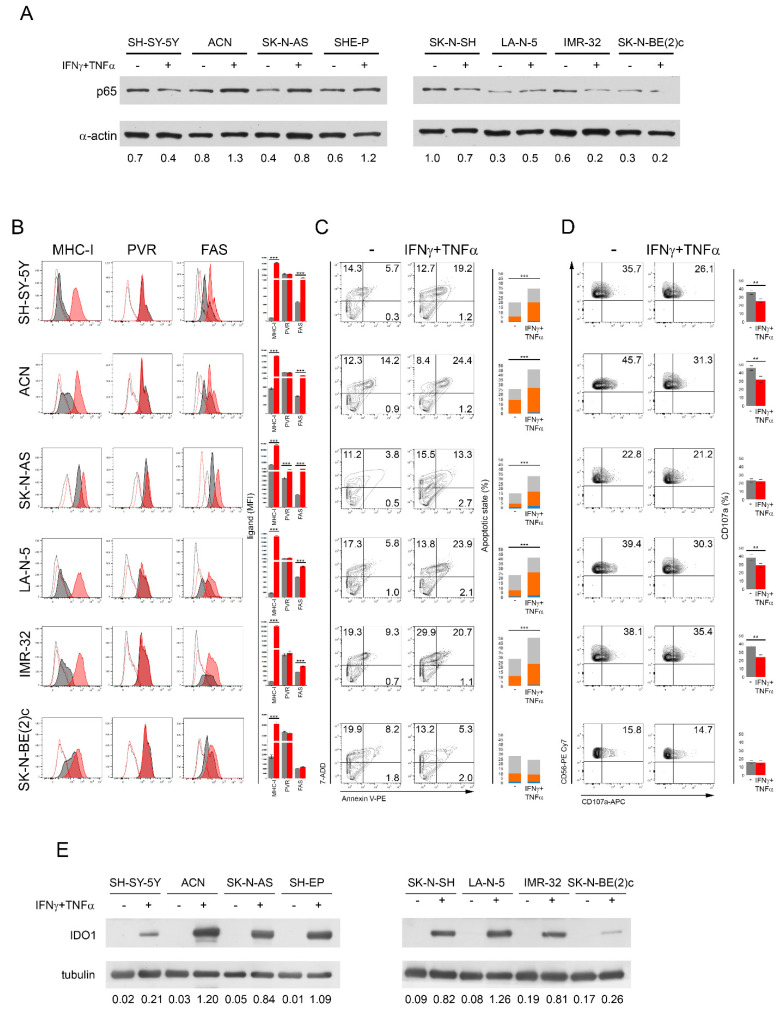
The treatment of NB cell lines with IFNγ and TNFα modulates p65 and increases Fas, thus rendering NB cells more susceptible to NK-cell-mediated apoptosis but not to NK cell degranulation. (**A**) NB cell lines were treated with IFNγ and TNFα for 48 h and analyzed for the expression of p65 by Western blotting. A representative of two independent experiments is reported. Below each panel, the normalized densitometry measured by ImageJ analysis is reported. (**B**) IFNγ and TNFα-treated NB cell lines were evaluated for the surface expression of MHC class I, PVR, and Fas by flow cytometry, together with the other ligands shown in Appendix A. Isotype-matched negative control antibodies are displayed as dashed gray and red lines for untreated and cytokine-treated NB cell lines, respectively. A representative experiment out of the four performed (left panel) and their summary reported in histograms (right panel) are shown; *** *p* < 0.001. (**C**) Untreated and cytokine-treated NB cells were used as a target in a NK-cell-mediated apoptotic assay. A representative experiment, out of four performed, for each tested NB cell line indicated (right panel) and their summary in stacked histograms (left panel) are reported. The percentage of cells in the early phase (Annexin V^+^ 7-ADD^−^, blue) and the late phase (Annexin V^+^ 7-ADD^+^, orange) of apoptosis as well as dead cells (Annexin V^−^ 7-ADD^+^, gray) are reported for each plot; *** *p* < 0.001. (**D**) Untreated and cytokine-treated NB cells were used as a target in a NK-cell-mediated degranulation assay. Degranulation of CD45^+^CD56^+^CD3^−^ NK cells, measured as CD107a cell-surface expression upon co-culture with the indicated NB cell lines untreated and treated with IFNγ and TNFα for 48 h. K562 target cells were used as a positive control (Appendix A). The percentage of CD107a^+^ NK cells is indicated in each plot. A representative experiment out of four performed (right panel) and their summary (left panel) are shown; ** *p* < 0.01. (**E**) NB cell lines untreated and treated with IFNγ and TNFα for 48 h were analyzed for the expression of IDO1 by Western blotting. A representative of two independent experiments is reported. Below each panel, the normalized densitometry with respect to tubulin expression used as a housekeeper, measured by ImageJ analysis, is reported. All original western blots are included in Appendix A.

**Figure 4 cancers-13-04368-f004:**
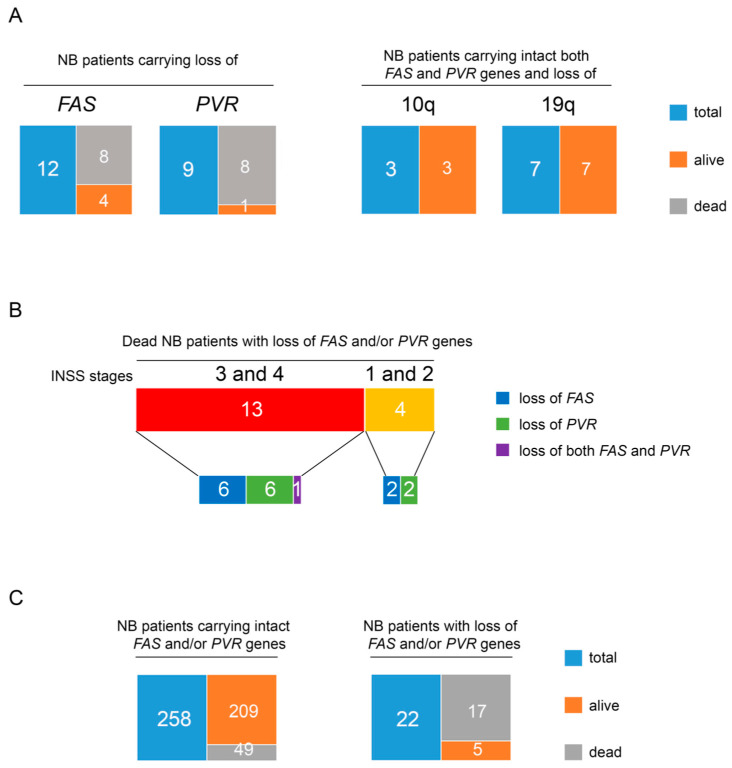
Loss of *FAS* and *PVR* in NB patients correlates with low survival regardless of disease stage. (**A**) Tree diagrams showing the number of alive and dead patients in NB case groups carrying loss of *FAS* or *PVR* genes (right panel) and those carrying both intact *FAS* and *PVR* genes and loss of chromosome 10q or 19q (left panel). (**B**) Tree diagrams showing the number of dead NB patients with loss of *FAS* or *PVR* or both genes at the INSS disease Stage 3 and 4 or 1 and 2. (**C**) Tree diagrams showing the number of alive and dead patients in the groups of NB cases carrying intact *FAS* and/or *PVR* genes (right panel) and those carrying loss of *FAS* and/or *PVR* genes (left panel).

**Table 1 cancers-13-04368-t001:** Status of *MYCN*, *FAS*, *PVR,* and *RELA* genes in NB cell lines.

NB Cell Lines	*MYCN* Gene	Chromosomal Coordinates of *MYCN* (2p24.3) Status	*FAS* Gene	Chromosomal Coordinates of *FAS* (10q23.31) Gain or Loss	*PVR* Gene	Chromosomal Coordinates of *PVR* (19q13.31) Gain or Loss	*RELA* Gene	Chromosomal Coordinates of *RELA* (11q13.1) Gain or Loss
SH-SY-5Y	single copy	_	single copy	_	single copy	_	single copy	_
ACN	single copy	_	single copy	_	gain	Chr19: 44,169,034–59,092,570 Cytoband: 19q13.31–q13.43 Size: 14.9 Mb	single copy	_
SK-N-AS	single copy	_	loss	Chr10: 43,615,122–135,474,787 Cytoband: 10q11.21–q26.3 Size: 91.8 Mb	single copy	_	single copy	_
SH-EP	single copy	_	gain	Chr10: 90,628,315–103,956,178 Cytoband: 10q23.31–q24.32 Size: 13.3 Mb	single copy	_	single copy	_
SK-N-SH	gain	Chr2: 17,019–48,571,447 Cytoband: 2p25.3–p16.3 Size: 48.5 Mb	single copy	_	single copy	_	single copy	
GICAN	gain	Chr2: 16,066,442–18,776,030 Cytoband: 2p24.3–p24.2 Size: 2.7 Mb	single copy	_	loss	Chr19: 27,764,285–50,857,784 Cytoband: 19q11–q13.33 Size: 23 Mb	gain	Chr11: 55,715,776–70,511,684 Cytoband: 11q12.1–q13.4 Size: 14.8 Mb
LA-N-5	amp	Chr2: 15,496,660–17,046,138 Cytoband: 2p24.3–p24.2 Size: 0.4 Mb	single copy	_	single copy	_	single copy	_
SMS-KCNR	amp	Chr2: 16,036,272–16,428,878 Cytoband: 2p24.3 Size: 1.5 Mb	single copy	_	single copy	_	single copy	
IMR-32	amp	Chr2: 14,773,079–16,086,291 Cytoband: 2p24.3 Size: 1.3 Mb	gain	Chr11: 85,989,063–134,446,160 Cytoband: 11q14.2–q25 Size: 48.4 Mb	single copy	_	single copy	_
SK-N-BE(2)c	amp	Chr2: 16,082,217–16,469,668 Cytoband: 2p24.3 Size: 0.4 Mb	loss	Chr10: 42,418,957–135,434,178 Cytoband: 11q11.21–q26.3 Size: 93 Mb	loss	Chr19: 27,853,207–59,057,101 Cytoband: 19q11–q13.43 Size: 31.2 Mb	loss	Chr11: 55,050,707–65,540,114 Cytoband: 11q11–q13.1 Size: 10.5 Mb
LA-N-1	amp	Chr2: 16,066,442–16,487,029 Cytoband: 2p24.3 Size: 0.4 Mb	loss	Chr11: 93,455,106–119,038,765 Cytoband: 11q21–q23.3 Size: 25.6 Mb	loss	Chr19: 36,685,449–59,092,570 Cytoband: 19q13.12–q13.43 Size: 22.4 Mb	loss	Chr3: 133,671,502–197,801,441 Cytoband: 3q22.1–q29 Size: 64.1 Mb

**Table 2 cancers-13-04368-t002:** Diagnostic characteristics of NB patients with tumors showing loss of 10q and/or loss of 19q.

Case n°	Age at Onset	INSS Stage	INRG Stage	Subtype	Differentiation Grade	*MYCN* Status	Cytoband and Chromosomal Coordinates of 10q Loss	Loss or cnLOH of *FAS* Gene (10q23.31)	Cytoband and Chromosomal Coordinates of 19q Loss	Loss of *PVR* Gene (19q13.31)	Relapse	Follow-Up	Disease State
1	2y 9m	3	L2	NB/GNBL	SD	amp	10q21.1–q26.3; chr10: 57,654,752–135,411,735	yes	19q12–q13.43; chr19: 30,367,571–59,057,101	yes	yes	dead	-
2	1y 10m	4	M	NB/GNBL	SD	amp	10q22.3–q26.3; chr10: 81,234,748–135,404,523	yes	-	no	yes	dead	-
3	3y 11m	4	M	NB/GNBL	SD	sc	10q23.1–q24.1; chr10: 82,434,763–98,098,105	yes	-	no	yes	dead	-
4	3y 9m	4	M	NB/GNBL	SD	sc	10q11.21–q26.3; chr10: 45,247,685–135,372,492	yes	-	no	yes	dead	-
5	0y 10 m	3	L2	NB NAS	SD	amp	10q21.3–q26.3; chr10: 64,979,620–135,404,523	yes	-	no	yes	dead	-
6	3y 1m	3	L2	NB/GNBL	SD	amp	10q11.21–q24.33; chr10: 43,615,579–105,634,097	yes	-	no	yes	dead	-
7	3y 10m	3	L2	NB/GNBL	SD	gain	10q11.22–q26.3; chr10: 48,334,407–135,404,523	yes	-	no	yes	dead	-
8	1y 2m	1	L1	NB/GNBL	SD	amp	10q22.1–q26.3; chr10: 73,406,500–135,404,523	yes	-	no	yes	dead	-
9	3y 1m	1	L1	NB/GNBL	SD	sc	10q22.3–q26.11; chr10: 78,015,106–121,458,431	yes	-	no	yes	dead	-
10	3y 4m	3	L2	NB/GNBL	SD	amp	10q11.23–q26.3; chr10: 50,955,699–135,372,492	yes	-	no	yes	alive	AD
11	5y 5m	3	L2	NB/GNBL	SD	sc	10q22.2–q24.33; chr10: 75,923,421–105,458,525	yes	-	no	no	alive	AD
12	16y 8m	2A	L1	NB/GNBL	SD	sc	10q11.21–q24.32; chr10: 42,969,765–103,988,947	yes	-	no	yes	alive	CR
13	4y 5m	2B	L2	NB/GNBL	nodular	gain	cnLOH10q11.23–q26.3; chr10: 4,541,520–67,615,559	yes	-	no	yes	alive	CR
14	4y 7m	4	M	NB/GNBL	SD	amp	10q23.32–q26.3; chr10: 93,204,607–135,404,523	no	19q13.12–q13.32; chr19: 38,263,097–47,823,090	yes	no	dead	-
15	2y 5m	4	M	NB/GNBL	SD	sc	-	no	19q13.31–q13.43; chr19: 49,797,610–59,092,570	yes	yes	dead	-
16	4y 4m	4	M	NB/GNBL	SD	gain	-	no	19q13.11–q13.32; chr19: 33,871,166–57,484,480	yes	yes	dead	-
17	4y 5m	4	M	NB/GNBL	SD	amp	-	no	19q13.2–q13.43; chr19: 41,157,895–58,940,734	yes	yes	dead	-
18	0y 7m	4	M	NB/GNBL	SD	amp	-	no	19q13.2–q13.33; chr19: 39,065,313–49,272,886	yes	yes	dead	-
19	0y 1m	3	L2	NB/GNBL	SD	sc	-	no	19q12–q13.43; chr19: 29,079,304–58,900,941	yes	yes	dead	-
20	6y 10m	2	L1	NB/GNBL	SD	sc	-	no	19q13.12–q13.32; chr19: 39,871,307–47,539,002	yes	yes	dead	-
21	1y 2m	1	L1	NB/GNBL	SD	amp	-	no	19q13.11–q13.43; chr19: 33,871,307–59,063,507	yes	yes	dead	-
22	5y 2m	4	M	NB/GNBL	SD	sc	-	no	19q13.31–q13.43; chr19: 49,797,610–59,057,101	yes	yes	alive	AD
23	3y 9m	4	M	NB/GNBL	SD	gain	10q21.1; chr10: 55,252,216–56,505,255	no	-	no	yes	alive	CR
24	4y 4m	4	M	NB/GNBL	SD	sc	10q11.22–q22.3; chr10: 49,797,866–77,817,731	no	-	no	no	alive	CR
25	1y 8m	4	M	NB/GNBL	SD	sc	-	no	19q13.32–q13.43; chr19: 46,527,590–59,063,507	no	no	alive	CR
26	0y 4m	4s	Ms	NB/GNBL	SD	sc	-	no	19q13.32–q13.43; chr19: 47,539,002–59,057,101	no	no	alive	CR
27	2y 10m	1	L1	NB/GNBL	SD	sc	10q11.21–21.1; chr10: 42,969,765–57,723,272	no	-	no	no	alive	CR
28	1y 2m	1	L1	NB/GNBL	SD	sc	-	no	19q13.32–q13.43; chr19: 47,273,347–59,057,101	no	no	alive	CR
29	8y 1m	1	L1	NB/GNBL	SD	sc	-	no	19q13.2–q13.31; chr19: 43,112,945–43,681,708	no	no	alive	CR
30	0y 4m	1	L1	NB/GNBL	SD	sc	-	no	19q13.32–q13.43; chr19: 51,784,149–63,407,936	no	no	alive	CR
31	0y 8m	1	L1	NB/GNBL	SD	sc	-	no	19q13.33–q13.41; chr19: 51,331,109–51,772,099	no	no	alive	CR
32	0y 11m	1	L1	NB/GNBL	SD	sc	-	no	19q13.32–q13.43; chr19: 46,527,590–58,940,734	no	no	alive	CR

**Table 3 cancers-13-04368-t003:** Association between *FAS* and/or *PVR* gene loss and fatal outcome in NB patients.

Patient Outcome	NB Patients Carrying Intact *FAS* and/or *PVR* Genes	NB Patients with Loss of *FAS* and/or *PVR* Genes
alive	209/258 (81%)	5/22 (23%)
dead	49/258 (19%)	17/22 (77%) *

* loss of *FAS* and/or *PVR* genes and alive vs dead patients: *p* < 0.00001; Fisher’s exact test.

## Data Availability

Data is contained within the article or Appendix A.

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
