# Peer review of "Enhancement of Neuroblastoma NK-Cell-Mediated Lysis through NF-kB p65 Subunit-Induced Expression of FAS and PVR, the Loss of Which Is Associated with Poor Patient Outcome"

_cancers, 2021, doi:10.3390/cancers13174368_

Round 1

Reviewer 1 Report

The authors have addressed my concerns to reaasonable extent.

Reviewer 2 Report

The authors have adequately addressed the major concerns of Reviewer #2.

This manuscript is a resubmission of an earlier submission. The following is a list of the peer review reports and author responses from that submission.

Round 1

Reviewer 1 Report

This manuscript claims that the upregulation of surface expression of Fas and PVR through transient NF-kB activation, as shown here with the overexpression of p65 NF-kB subunit, holds promise as a therapeutic modality for enhancing the susceptibility of neuroblastoma (NB) to NK cell-mediated lysis. They used various NB cell lines with different MYCN gene status (single copy, gain, amplification) and showed that the upregulation of Fas and PVR was more evident in MYCN non-amplified NB cell lines. Furthermore, they found that loss of FAS and/or PVR gene correlated with low survival using array CGH analysis in a cohort of NB patients. Overall, the study is largely correlative but lacks the direct evidence to support the importance of NF-kB in the upregulation of Fas and PVR and patient prognosis by their results presented.

Specific comments

It appears that the overexpression of p65 in NB cells is not practical therapeutic strategy for NK cell-based therapy. Moreover, the role of NF-kB in the upregulation of Fas and PVR is already known in other cancer cells. In this respect, it would be better to show a relevant example of upregulation of Fas and PVR and consequent increased NK cell susceptibility via NF-kB activation with some molecules (i.e. drug) being considered for NB therapy.

They claimed a negative role of MYCN in the p65-mediated induction of Fas and PVR using a MYCN single copy cells (SH-SY-5Y) in comparison to three MYCN amplified cells (LA-N-5, SMS-KCNR, and SK-N-BE(2)c). To support their claim, they should use more MYCN non-amplified cells for such comparison.

Figure 1A: It is required to show a representative FACS profile of surface expression of MHC class I along with the indicated ligands for NK activating receptors in the main or supplementary figures.

Figure 2: To directly correlate Fas and PVR upregulation with increased NK susceptibility via NF-kB activation, they should use antibody-mediated blockade of Fas and PVR receptors in p65-overexpressed NB cells and/or specific NF-kB activator in parental NB cells.

It is hard to understand the description of array CGH analysis in the results and Table 2. It would be better to present the results with a figure showing a correlation among MYCN, Fas, PVR, disease state, and more desirably NF-kB expression level.

Author Response

Reviewer 1

Comments and Suggestions for Authors

This manuscript claims that the upregulation of surface expression of Fas and PVR through transient NF-kB activation, as shown here with the overexpression of p65 NF-kB subunit, holds promise as a therapeutic modality for enhancing the susceptibility of neuroblastoma (NB) to NK cell-mediated lysis. They used various NB cell lines with different MYCN gene status (single copy, gain, amplification) and showed that the upregulation of Fas and PVR was more evident in MYCN non-amplified NB cell lines. Furthermore, they found that loss of FAS and/or PVR gene correlated with low survival using array CGH analysis in a cohort of NB patients. Overall, the study is largely correlative but lacks the direct evidence to support the importance of NF-kB in the upregulation of Fas and PVR and patient prognosis by their results presented.

Specific comments

It appears that the overexpression of p65 in NB cells is not practical therapeutic strategy for NK cell-based therapy. Moreover, the role of NF-kB in the upregulation of Fas and PVR is already known in other cancer cells. In this respect, it would be better to show a relevant example of upregulation of Fas and PVR and consequent increased NK cell susceptibility via NF-kB activation with some molecules (i.e. drug) being considered for NB therapy.

R: We thank the Reviewer for raising this point. In the search for clinical approaches that lead to the upregulation of Fas and PVR, thereby rendering NB cells more susceptible to NK cell mediated-killing, we tested the efficacy of two cytokines IFNg and TNFa, known to upregulate Fas in different contexts through the activation of NF-Kb. We treated 5 MYCN non-amplified NB cell lines such as SH-SY-5Y, ACN, SK-N-AS, SHE-P and SK-N-SH and 3 MYCN amplified NB cell lines such as LA-N-5, IMR-32 and SK-N-BE(2)c NB cell lines with IFNg and TNFa for 48 hours. The cytokine treatment induced a modulation of p65 expression level with an increase in ACN, SK-N-AS, SHE-P, and, to a lesser extent, LA-N-5 cell lines (new Figure 3A). Interestingly, the IFNg and TNFa treatment significantly increased the expression of Fas in all NB cell lines with the exclusion of SK-N-BE(2)c used as negative control, but of PVR only in SK-N-AS cell line (new Figure 3B). In addition, the cytokine treatment induced the increase of ULBP1 in SH-SY-5Y, SK-N-AS and IMR-32, of ULBP3 in LA-N-5, and of Nectin-2 in IMR-32 (new Supplementary Figure 4A). By contrast, the cytokine treatment induced the downregulation of ULBP2/5/6 in SH-SY-5Y and LA-N-5, of ULBP-3 in SH-SY-5Y, of Nectin-2 in SH-SY-5Y, ACN and LA-N-5, and of TRAIL-R2 in all NB cell lines (new Supplementary Figure 4A). As expected, MHC class I expression was strongly upregulated by the cytokine treatment in all NB cell lines (new Figure 3B). Accordingly, all NB cell lines treated with IFNg and TNFa (with the exclusion of SK-N-BE(2)c) were significantly more susceptible to NK cell-mediated apoptosis than untreated cells (new Figure 3C), whereas the cytokine treatment alone did not affect the apoptotic status of NB cell lines (new Supplementary Figure 4B). In contrast, the NK cell-mediated degranulation in response to IFNg- and TNFa-treated NB cells was significantly reduced, except for SK-N-AS and SK-N-BE(2)c in which it was unchanged. These data suggest that the downregulation of ligands for NK cell-activating receptors (new Supplementary Figure 4A) as well as the strong upregulation of MHC class I upon cytokine treatment contributed to the impaired NK cell-degranulation in response to cytokine-treated NB cell lines.

Furthermore, to explore further mechanisms of immune evasion leading to the impaired NK cell-degranulation in response to cytokine treated-NB cells, we assessed the expression level of Indoleamine-pyrrole 2,3-dioxygenase1 (IDO1), an enzyme known to be involved in mechanisms conferring resistance to immune cell activities in most tumors, including NB. All NB cell lines treated with IFNg and TNFa showed a strong upregulation of IDO1 (new Figure 3D), indicating that the treatment with these cytokines, although it resulted in an increased Fas expression and consequently enhanced NK cell-mediated apoptosis of NB cell lines, controlled the NK cell-mediated recognition process through IDO1-dependent immune evasion mechanisms, thereby attenuating the NK cell-mediated killing. These data suggested that, to induce Fas and activating ligands such as PVR on NB cells, translational approaches based on the use of cytokines should be reviewed and the combined use of IDO1 inhibitors should optimize the NK cell-mediated killing of cytokine-treated NB cells.

They claimed a negative role of MYCN in the p65-mediated induction of Fas and PVR using a MYCN single copy cells (SH-SY-5Y) in comparison to three MYCN amplified cells (LA-N-5, SMS-KCNR, and SK-N-BE(2)c). To support their claim, they should use more MYCN non-amplified cells for such comparison.

R: In the revised manuscript and new Figure 1 we show data obtained by 2 MYCN non-amplified NB cell lines such as SH-SY-5Y and SK-N-AS and 4 MYCN-amplified NB cell lines such LA-N-5, SMS-KCNR, IMR-32 and SK-N-BE(2)c (new Figure 1). The stronger upregulation of Fas, although not obtained in RELA lost SK-N-AS by p65 over-expression, was particularly evident in the cytokine treatment experiments showing that MYCN non-amplified NB cells were more sensitive to Fas induction compared to MYCN-amplified NB cell lines (new Figure 3B). In contrast, the stronger upregulation of PVR following the p65 overexpression was evident in both MYCN non-amplified NB cell compared to MYCN-amplified NB cell lines (new Figure 1).

Figure 1A: It is required to show a representative FACS profile of surface expression of MHC class I along with the indicated ligands for NK activating receptors in the main or supplementary figures.

R: The new Figure 1 and new Supplementary Figure 2 show a representative FACS profile of the surface expression levels of all molecule tested

Figure 2: To directly correlate Fas and PVR upregulation with increased NK susceptibility via NF-kB activation, they should use antibody-mediated blockade of Fas and PVR receptors in p65-overexpressed NB cells and/or specific NF-kB activator in parental NB cells.

R: We thank the reviewer for this comment. We performed blocking experiments by using anti-DNAM-1 and anti-FasL indicating that both Fas and PVR are involved in the enhanced NK cell-mediated apoptosis and degranulation in response to p65 overexpressed NB cell lines (new Figure 2A-D).

It is hard to understand the description of array CGH analysis in the results and Table 2. It would be better to present the results with a figure showing a correlation among MYCN, Fas, PVR, disease state, and more desirably NF-kB expression level.

R: We thank the reviewer for this suggestion. In order to improve the messages contained in Table 2, we performed a new Figure 4 showing the outcome of our NB patients through tree diagrams, in relation to the status of FAS and PVR genes and patients disease stages. In addition, we added the status of RELA in Supplementary Table 1.

Reviewer 2 Report

(1). Overall I think significantly more data are needed to support the conclusion on the involvement of FAS and PVR in determining the neuroblastoma sensitivity to NK cell killing. The authors started the study by analyzing the gene copies, e.g., FAS and PVR, of 11 NB cell lines. I think it’s important to perform the cell line profiling analysis at the protein level, and also correlate the surface expressions of the various ligands as shown in Figure 1A with the sensitivity to NK cell killing for all the 11 NB cell lines. This will provide a much better foundation to the hypothesis that the authors proposed. I suggest using flow cytometry to first measure the panel of ligands as shown in Figure 1A for all 11 NB cell lines, and at the same time measuring the extent of NK cell killing of the 11 NB cell lines with killing assay shown in Figure 2. Correlation of the ligand expression and cell death response to NK cells cross the 11 NB cell lines shall reveal whether FAS and PVR indeed correlate with sensitivity to NK cell killing. I think this data is very important in justifying the overall study and should be shown as Figure 1. The arguments that the authors provided on selecting a few NB cell lines for further FACS and NK cell killing analysis are not convincing.  

(2). Target cell susceptibility to NK cell killing is known to be regulated by collective signaling from multiple inhibitory and activating NK cell receptors. I don’t think the authors presented unequivocal data that support FAS and PVR are particularly relevant for neuroblastoma. I think it’s important to perform additional functional assays, for example, using neutralizing antibodies against FAS or PVR to block their activities and see whether loss of their activities indeed attenuates NK cell killing both in control condition and in cells transfected with p65 subunit.

(3). The enhancement of NK cell-mediated target cell death upon p65 subunit transfection as shown in Figure 2D and 2F is only moderate, i.e., about 10-15% increase. It is not convincing that such moderate change in target cell sensitivity to NK cell killing is the key to modulate NK cell response of neuroblastoma either in vitro or in vivo. 

(4). Data of Table 2 are difficult to comprehend and interpret. Please organize and plot the data in figures for more intuitive interpretation. For example, you can correlate the disease state and the gene status in scatter plot. That shall be a lot easier to see the correlation pattern. 

Author Response

Reviewer 2

Comments and Suggestions for Authors

(1). Overall I think significantly more data are needed to support the conclusion on the involvement of FAS and PVR in determining the neuroblastoma sensitivity to NK cell killing. The authors started the study by analyzing the gene copies, e.g., FAS and PVR, of 11 NB cell lines. I think it’s important to perform the cell line profiling analysis at the protein level, and also correlate the surface expressions of the various ligands as shown in Figure 1A with the sensitivity to NK cell killing for all the 11 NB cell lines. This will provide a much better foundation to the hypothesis that the authors proposed. I suggest using flow cytometry to first measure the panel of ligands as shown in Figure 1A for all 11 NB cell lines, and at the same time measuring the extent of NK cell killing of the 11 NB cell lines with killing assay shown in Figure 2. Correlation of the ligand expression and cell death response to NK cells cross the 11 NB cell lines shall reveal whether FAS and PVR indeed correlate with sensitivity to NK cell killing. I think this data is very important in justifying the overall study and should be shown as Figure 1. The arguments that the authors provided on selecting a few NB cell lines for further FACS and NK cell killing analysis are not convincing.

R: Overall, in the revised version of our manuscript and figures we add the FAS and PVR profile for 6 NB cell lines and 5 for NK cell function, performed at least 4 times for statistical evaluation, compared with the data shown in the first version. We also obtained the same data with some other NB cell lines tested (SK-N-SH, SH-EP and LA-N-1) but the low number of experiments performed for reasons related to long cell culture times and very limited publication time, did not allow us to include them in this manuscript.  

(2). Target cell susceptibility to NK cell killing is known to be regulated by collective signaling from multiple inhibitory and activating NK cell receptors. I don’t think the authors presented unequivocal data that support FAS and PVR are particularly relevant for neuroblastoma. I think it’s important to perform additional functional assays, for example, using neutralizing antibodies against FAS or PVR to block their activities and see whether loss of their activities indeed attenuates NK cell killing both in control condition and in cells transfected with p65 subunit.

R: We thank the reviewer for this comment. We performed blocking experiments by using anti-DNAM-1 and anti-FasL indicating that both Fas and PVR are involved in the enhanced NK cell-mediated apoptosis and degranulation in response to p65 overexpressed NB cell lines (new Figure 2A-D).

(3). The enhancement of NK cell-mediated target cell death upon p65 subunit transfection as shown in Figure 2D and 2F is only moderate, i.e., about 10-15% increase. It is not convincing that such moderate change in target cell sensitivity to NK cell killing is the key to modulate NK cell response of neuroblastoma either in vitro or in vivo. 

R: The induction of NK cell functions observed in this set of experiments is consistent with those previously reported shown in our articles and those produced by other authors. Although it might seem moderate, it should be considerate that the degranulation and killing process measured by this assays reflects a situation that occurs within the first 3-4 hours of NK cell binding to target cells. It is not unusual to observe total target dead at times longer of coculture as at 16 hours after.

(4). Data of Table 2 are difficult to comprehend and interpret. Please organize and plot the data in figures for more intuitive interpretation. For example, you can correlate the disease state and the gene status in scatter plot. That shall be a lot easier to see the correlation pattern. 

R: We thank the reviewer for this suggestion. In order to improve the messages contained in Table 2, we performed a new Figure 4 showing the outcome of our NB patients through tree diagrams, in relation to the status of FAS and PVR genes and patients disease stages. In addition, we added the status of RELA in Supplementary Table 1.